# Spatial Spectral Band Selection for Enhanced Hyperspectral Remote Sensing Classification Applications

**DOI:** 10.3390/jimaging6090087

**Published:** 2020-08-31

**Authors:** Ruben Moya Torres, Peter W.T. Yuen, Changfeng Yuan, Johathan Piper, Chris McCullough, Peter Godfree

**Affiliations:** 1Department of Electronic Warfare, Cranfield University, Swindon SN6 8LA, UK; r.moya-torres@cranfield.ac.uk (R.M.T.); ycf1028@163.com (C.Y.); j.piper@cranfield.ac.uk (J.P.); bbop1983@live.com (C.M.); 2College of Transportation Management, Dalian Maritime University, Dalian 116026, China; 3Defence Science and Technology Laboratory (DSTL), Space and Sensing Systems Group, Porton Down, Salisbury, Wiltshire SP4 0JQ, UK; psgodfree@dstl.gov.uk

**Keywords:** band selection, spatial spectral band selection, hyperspectral imaging, classification, mutual information, curse of dimensionality, Hughes phenomenon, accuracy-dimensionality characteristics

## Abstract

Despite the numerous band selection (BS) algorithms reported in the field, most if not all have exhibited maximal accuracy when more spectral bands are utilized for classification. This apparently disagrees with the theoretical model of the ‘curse of dimensionality’ phenomenon, without apparent explanations. If it were true, then BS would be deemed as an academic piece of research without real benefits to practical applications. This paper presents a spatial spectral mutual information (SSMI) BS scheme that utilizes a spatial feature extraction technique as a preprocessing step, followed by the clustering of the mutual information (MI) of spectral bands for enhancing the efficiency of the BS. Through the SSMI BS scheme, a sharp ’bell’-shaped accuracy-dimensionality characteristic that peaks at about 20 bands has been observed for the very first time. The performance of the proposed SSMI BS scheme has been validated through 6 hyperspectral imaging (HSI) datasets (Indian Pines, Botswana, Barrax, Pavia University, Salinas, and Kennedy Space Center (KSC)), and its classification accuracy is shown to be approximately 10% better than seven state-of-the-art BS schemes (Saliency, HyperBS, SLN, OCF, FDPC, ISSC, and Convolution Neural Network (CNN)). The present result confirms that the high efficiency of the BS scheme is essentially important to observe and validate the Hughes’ phenomenon in the analysis of HSI data. Experiments also show that the classification accuracy can be affected by as much as approximately 10% when a single ‘crucial’ band is included or missed out for classification.

## 1. Introduction

Hyperspectral imaging (HSI) that exploits both spectral and spatial features of the scene [1,2], has made it a powerful technique for applications such as geographical mapping [3], classifications [4], and target detections [5,6], in multidisciplinary fields of agricultural [7], food industry [8], medical [9], and security [10], sectors. The usefulness of HSI mainly stems from the very detailed spectral information of the scene that it provides; however, it is also one of the drawbacks of HSI for achieving a high degree of classification or detection accuracy when it has high spectral dimension. For a model with a high dimensional (spectral) feature space, it will require a very high number of training samples to train the model properly due to the high degree of freedom in the model according to Hughes [1]. This is so-called ‘curse of dimensionality’ that manifests itself by the presence of a ‘bell’ shaped accuracy-dimensionality relationship of the classification model when it is trained under a limited amount of data. This means that the theoretical accuracy limit of the model will be reduced as the dimensionality of the data is increased further. One common way to rectify this drawback is the introduction of an effective band selection (BS) scheme, wherein a model is built such that a few image bands that contain rich information in the context of the objective of the task are selected or extracted from the scene for further processing. Due to the advantage of BS that enhances the end-to-end throughput of HSI significantly, a great many different schemes of BS have been proposed in the past couple decades within the HSI community [2,11,12,13,14,15,16].

The common goal amongst all these various streams of BS algorithm is to provide a robust extraction of the essential information of the scene. Approaches such as the maximization of spectral mutual information [17,18,19,20,21,22,23,24,25,26,27,28,29], decomposition of spectral information through sparsity [30,31,32,33,34,35], spectral unmixing [36,37,38], or related techniques such as variance-based optimizations [39,40,41,42,43,44,45,46,47,48,49,50,51,52], have been widely studied in the last two decades. Recent concepts that utilize both spatial and spectral information have shown improved classification performance over the spectral centric of BS schemes [53,54,55]. There are various different methodologies for implementing these concepts in BS: Clustering based [13,56,57,58,59,60,61,62,63,64], deep learning methods [65,66,67,68,69,70], machine learning methods [71,72,73,74], and a hierarchy of several methodologies combined together [75,76], have been widely reported. Furthermore, some of these require supervision in which training data are needed to optimize the model [77,78], while others are unsupervised without the need of prior information. One of the most outstanding problems in BS is the determination of the most appropriate number of bands that are needed to optimize the performance of the classification/detection task. Many algorithms require the user to specify the desired dimension, and the work on the virtual dimensionality [79], is still a matter of intensive research.

Despite the numerous amount of BS research in the field [2,3,4,5,6,7,8,9,10,11,12,13], the classification performances of these reported results are all exhibiting a ‘knee’-shaped accuracy-dimensionality curve, i.e., the accuracy of the classification increases steadily when more bands are utilized for the classification. Moreover, existing BS algorithms are all showing a maximum classification accuracy when all the spectral bands are utilized. These results apparently suggest that Hughes’ prediction may not be correct, or alternatively, there may be factors that have prevented this ‘bell’-shaped accuracy-dimensionality characteristic curve to be observed. (i) The number of training data is abundant enough to train the very high degree of freedom of the classification model sufficiently. (ii) The reported BS schemes are not efficient enough to reveal the intrinsic (i.e., theoretical) accuracy-dimensionality characteristics. (iii) An ‘ideal’ dataset in which all bands possess an equal extent of information has been assumed in the Hughes theoretical modeling; however, this ideal dataset is never realized in the real world. Hence, the shape of the accuracy-dimensionality plot of the real world data is heavily dependent on how the bands are selected; for example, the successive selection of more bands will enhance the accuracy only when the selected bands are highly informative. (iv) Other factors such as the Fisher criterion, which arises from issues of classes overlapping, Cover′s reasoning, which concerns with the number of training size with respected to the dimensionality of the dataset, and other imaging artefacts such as sensor noise, spectral mixing, etc., which may also prevent the Hughes phenomenon from being observed from the real-world data.

To ensure that the last factor does not dominate the classification accuracy characteristic, six widely studied publicly available HSI datasets have been adopted here, and the result is subsequently compared with respect to 7 other state-of-the-art BS algorithms. To maintain the integrity of the dataset as much as possible (see Section 2.1 for more information), a spatial-based preprocessing technique is applied here to assign all non-informative bands (noisy and/or featureless bands) to the bottom (i.e., least) priority list for the band selection. One objective of this work is to propose a spatial spectral band selection scheme for enhancing the classification efficiency and with a view to understanding the origin of why the curse of dimensionality phenomenon is so difficult to observe in the real-world data. The focus of the present paper is to address factors (ii) and (iii) above; hence, a BS scheme that utilizes both spatial and spectral information for enhancing the classification accuracy of hyperspectral imagery has been designed. The BS scheme utilizes a spatial feature extraction as a preprocessing step, followed by a basic mutual information (MI) spectral feature-based BS. This is named as a spatial spectral mutual information (SSMI) scheme, and its performance is then compared directly with the basic MI BS.

The layout of the paper is as follows. Section 1 outlines the background of the work related to the topic of band selection, and the motive/objective of the research are outlined. Section 2 describes the principles and algorithm of the proposed SSMI method together with an outline of the working principles of seven state-of-the-art competing BS algorithms. Then, the datasets to be studied in this work and the assessment metric are briefly presented. Section 3 presents the classification performance of the proposed SSMI and the seven competing BS algorithms over six HSI datasets. Section 4 presents a discussion of how sensitive the classification accuracy is when ‘crucial’ bands are added or omitted. Section 5 concludes the paper by highlighting the importance of the efficiency of band selection schemes and how can it be assessed more meaningfully in future research directions.

## 2. Methods and Materials

### 2.1. Spatial Preprocessing Method

The main concept in band selection is to establish the methodology to allow the most informative bands in the dataset to be extracted for further processing. Informative bands in the context of land-use classification applications are those that contain rich features within the band to allow different types of objects to be discriminated throughout the scene. In spatial terms, the informative bands are those that are rich in morphological features. There are numerous ways to achieve this objective for the selection of highly informative bands from the spatial perspective, e.g., through the similarities of neighborhood pixels [54], the construction of a matrix-based margin-maximization using the local spatial pixel neighborhood information [55], and also the extraction of saliency spatial feature through a simplistic edge detection methodology [80].

Inspired by the simple yet effective methodology by using edge detections for the extraction of morphology features from each band [80], this concept has been adopted in this work as the spatial preprocessing technique. The objective of this spatial preprocessing is to remove spectral bands that are low in morphological features, i.e., non-discriminative bands. Given a hyperspectral image that consists of N bands B = (B_1_, B_2_, …, B_N_), the morphology of the bands can be approximated by using edge detection:E_i_ = Edge (B_i_)(1)
where Ei is the edge feature map of the band Bi, and Edge(.) is the edge detector operator, which can be Canny detectors, Sobel detectors, etc. There are various options to classify the edge feature maps into highly structured maps and distinguish them from the featureless low morphological maps. One straightforward means is to rank the Ei by comparing it with respect to the mean of all Ei:C_i_ = Corr (E_i_, E)
and
E = mean(Ei)(2)

The Corr(.) is the correlation function, and an abrupt change of Ci will give an indication of the threshold boundary between the highly structured and the featureless morphological maps. In this work, all bands below this threshold are reassigned to the bottom of the selection list, and the highly structured bands are then passed onto the next stage of spectral processing. The validity of this formulation is more applicable to high-quality datasets (i.e., high signal-to-noise ratio) of scenes with a very small portion of low reflectance objects such as water or hard shadows.

### 2.2. Spectral Band Selections Using Mutual Information (MI) 

Entropy has been a quantity that has been widely used in communication, computing, cryptography, and many other data-related applications. Entropy is a measure of the unpredictability of the state, so it is not only the content of the state but also how the state is chosen that determines its entropy. Given a variable *K* and that the probability of the event *K_i_* is p(*K_i_*), the amount of information I(*K_i_*) acquired due to the observation of the event *K_i_* according to Shannon [81], is defined as:(3)$$I\left(K_{i}\right)=\log\left(\frac{1}{p\left(K_{i}\right)}\right)=-\log(p\left(K_{i}\right)$$
The entropy (H) of the variable *K* for $\forall K,\;K\in [K_{1}\ldots K_{i}\ldots K_{n}],$ (represented by H(*K*)) is the expectation value of the information function I(*K*), i.e., the expectation of the probability (p(*K*)) for choosing elements of *K*:(4)$$H\left(K\right)=\;-\underset{i}{\sum }p\left(K_{i}\right)\;\log\;p\left(K_{i}\right)\;$$

A large entropy H(A) means that A is very unpredictable, and the averaged amount of the information conveyed by the identification of A is large. In band selection (BS), the band image can be considered as A, and the pixels in the band are represented by A_i_. Thus, the entropy can be used for encoding the information of every band in the dataset by using Equation (4) [21,22,23,24,25,26,27,28,29]. An alternative methodology has been using joint entropy between two variables A and B [17,82,83,84,85]. From Equation (3), the joint information of variables A and B is given by the mutual information (MI) I(A,B), which is in the form of:$$I\left(A,B\right)=\;\underset{A,\;B}{\sum }p\left(A,B\right)\log\frac{p\left(A,B\right)}{p\left(A\right)\;\times \;p\left(B\right)}\;$$
(5)I(A,B)= H(A)+H(B)−H(A, B)
where p(A) and p(B) are the marginal probability distributions of variables A and B, respectively, and p(A, B) is the joint probability distribution of variables A and B. Equation (5) implies that the I (A,B) can be used to measure the similarity of two variables A and B. If A and B are two spectral bands in the image, the I(A, B) measures the independency (or similarity) of (the information convey in) these two bands. In the case when one variable is chosen to be the reference data, and the other variable is the band images of the scene, then the joint information of I(A,B) measures how close (in the context of information) the band images are with respect to the reference data such as the labeled data that have been used previously [17].

### 2.3. Spatial Spectral Mutual Information Band Selections (SSMI) 

The spatial spectral band selections method that is to be utilized in this work is the combination of the spatial preprocessing outlined in Section 2.1 and it cascades the output into the spectral band selection using mutual information as discussed in Section 2.2. In this paper, the marginal probabilities of the bands are estimated from the normalized band pairs, and their corresponding entropies are then calculated by using Equation (4). The joint entropy is subsequently evaluated for each pair of adjacent image bands, and the mutual information (MI) of the image pair (i.e., the I(A, B)) is evaluated according to Equation (5). Then, the numerous presentation (i.e., the value) of the I(A, B) (i.e., Mutual Information) for all band pairs of the dataset are ranked in ascending order, and the topmost ones represent the most informative bands, with all noisy/featureless bands at the bottom of the list. Then, this selection list is utilized for the BS. This algorithm is termed as spatial spectral mutual information (SSMI), and the pseudo-code of the proposed method is outlined in the Algorithm 1 (see Table 1 for more information).

### 2.4. Competing Band Selection Algorithms

In this study, seven state-of-the-art BS algorithms have been selected from the literature for a direct comparison with the classification performance of the proposed SSMI algorithm. The working principles of these algorithms are briefly outlined in the following subsections to let the readers understand the difference between these competing methods with respect to the one that we propose here.

#### 2.4.1. Saliency Bands and Scale Selection (SBSS) 

The SBSS (Saliency Bands and Scale Selection) [86], method utilizes both spatial and spectral information for its band selection. The principle of the method is to identify the saliency of each band through the numbers of the extrema points of the Hessian matrix of the band image.

#### 2.4.2. HyperBS

This algorithm utilizes the correlation relationship of bands to split or merge according to their mutual correlations [87]. With a user-defined threshold, spectral bands are split when the correlation of a pair of bands are below the threshold. Otherwise, they will be merged to reduce the dimensionality of the hyperspectral image.

#### 2.4.3. SLN (Single-Layer Neural Networks)

This is a neural network-based algorithm [88], which assigns the weight of the neurons according to the correlations of the cross-entropy of the bands. One advantage of this method is that the bands with the highest and lowest weight values are selected, and the selection is class dependent.

#### 2.4.4. OCF (Optimal Clustering Framework)

The OCF [13], is a framework that is designed to extract the abundance of the contributions from the bands toward different classes of the image data. The framework involves an arbitrary clustering method such as K-Means, which clusters the input image into an arbitrary number of classes to initiate the process. Then, the contributions of the bands toward each cluster are evaluated and they are then clustered into groups, and the final selected bands are extracted from these band contribution clusters.

#### 2.4.5. E-FDPC (Enhanced Fast Density-Peak-Based Clustering)

The E-FDPC [61], method is based on a cluster ranking principle similar to that of the OCF technique. It is known that the points at the cluster center possess the largest local density (i.e., high number of points) and intra-cluster distances. FDPC uses this fact to locate the center of clusters of spectral bands through ranking the similarity matrix of band pair according to their products of the local densities and the intra-cluster distances. The FDPC method is highly empirical, and the enhanced version E-FDPC updates the local density function to eliminate tuning parameters. The selected bands are those at the top of the rank, thus excluding the highly correlated bands that are at the bottom of the list.

#### 2.4.6. ISSC (Improved Sparse Subspace Clustering)

The ISSC [32], makes use of the compressive sensing technique, which has been widely used for finding the atoms (i.e., endmembers) of the dictionary. This method finds a few atoms in the spectral domain such that other bands can be reconstructed through a linear combination of these ‘spectral’ atoms. As in compressive sensing, the atoms are found by minimizing the L2 norm of the reconstructed bands with respect to the raw data. Then, a similarity matrix between a pair of sparse band vectors is constructed through the sparse coefficients of both vectors. Then, the similarity matrix is segmented into clusters, and the bands that are closest to the center of the clusters are chosen as the most informative bands. The concept is similar to both OCF and E-FDPC but different in methodology.

#### 2.4.7. CNN (Convolutional Neural Network)

This band selection method [89], is based on a 5-layer CNN network that consists of 3 convolution layers, 2 fully connected layers, and a final softmax layer. The network is firstly trained through a substantial amount of band images as training data. Subsequently, it extracts features from each test band image, and the weight of the neurons is updated depending on whether the predicted label is correctly or incorrectly classified. Then, the network combines all models into an ensemble, and the mostly weighted bands are selected by voting.

### 2.5. HSI Datasets Employed in This Paper

Six widely studied publicly available datasets—namely, the Pavia University, Indian Pines, Barrax, Salinas, Kennedy Space Center, and Botswana [90], have been employed in this work for the validation of the proposed SSMI BS method. The Barrax dataset was acquired during the 1999–2006 VALERI campaigns [91]. This dataset was acquired by the DAIS sensor over the 5 km × 5 km Barrax site in Albacete, Spain. It consists of 400 × 400 pixels and 128 bands with 18 classes of vegetation and crops [91,92]. The Pavia University hyperspectral dataset was acquired by the ROSIS sensor during a flight campaign over Pavia, northern Italy. The ground sampling distance (GSD) is 1.3 m, and the dataset dimension is 340 × 610 pixels with 103 bands. The Indian pines dataset has been one of the most widely studied imagery in the remote sensing research. It was acquired by the AVIRIS (Airborne Visible/Infrared Imaging Spectrometer) sensor, and the imagery contains 145 × 145 pixels with 224 bands ranging from 400 to 2500 nm. There are 4 bad bands and low signal to noise (SNR) bands due to water absorptions such as those between (104–108), (150–163), and also the band 220: They have all been removed before the data analysis. The Salinas scene was collected by the 224-band AVIRIS (Airborne Visible/Infrared Imaging Spectrometer) sensor over Salinas Valley, California, and it is characterized by high spatial resolution of GSD 3.7 m with 86 lines and 83 samples. Similar to that of the Indian Pines image, 20 water absorption bands between the (108–112), bands, (154–167), bands, and the 224 band have been removed, leaving 204 bands for data analysis. The Kennedy Space Center (KSC) dataset was acquired by the AVIRIS sensor in Florida on the 23 March 1996. The imagery was acquired at high altitude of 20 km with a GSD of 18 m. The dataset consists of 521 × 614 pixels in 224 bands covering a spectral region of 400–2500 nm with a narrow 10 nm FWHM per band. After removing water absorption and low SNR bands, there are 176 bands remaining for data analysis [93]. The Botswana dataset was acquired by the Hyperion sensor (EO-1) on the 31 May 2001, and it has dimensions of 1476 × 256 pixels in 220 bands. The GSD of Botswana is 30 m, which is the least spatial resolution over the other 5 datasets that have been employed in this study. After removing noisy and water absorption bands, there are 145 good quality bands remaining for data analysis [94,95].

The pseudo-RGB picture and the ground truth (GT) classification of these 6 datasets are presented in Figure 1, and the class information such as the size and nature of each class is tabulated in Table 2. It is noted that all datasets have reasonable class sizes *except* for the Botswana, which has an average class size of 232. The standard deviation of class sizes in Botswana is 67, which is 3 times smaller than that of the Indian Pines (STD = 650). In the Botswana data, all class sizes are relatively uniform, which can be seen from the GT map shown in Figure 1d. In other words, the averaged overall accuracy (OA) of this dataset will give a better indication of how the band selections affects the classification performance.

### 2.6. Experimental Configuration and Metrics for Assessing Classification Performances

Throughout this work, all experimental runs were repeated five times to obtain an average of the accuracy and standard deviations of the classification results. Most of the experimental runs used 10% of training data per class, and the classification was performed using SVM and also KNN.

In this study, the overall accuracy (OA) and the Kappa coefficient have been adopted as the assessment metrics for the indication of the classification performance as the result of band selections. The OA is calculated by the sum of the correctly classified pixels from each class and the ratio of this sum with the total number of pixels of all classes in the reference GT map. Thus, the OA can be skewed by ‘easy’ targets or/and large class sizes in the dataset. The kappa statistic is a measure of the overall agreement of the classification accuracy. It is used to control the instances that may have been correctly classified by chance. This can be calculated using both the observed (total) accuracy and the random accuracy: Kappa = (total accuracy − random accuracy)/(1 − random accuracy).

## 3. Results 

### 3.1. Band Selection (BS) Using Spectral Information Only

As outlined in Section 2, band selection (BS) using spectral information alone may not be able to extract essence information from hyperspectral data effectively. Figure 2 depicts the classification result by SVM using two typical classical BS schemes: (a) mutual information (MI)-based (see Section 2.2 for more information) [17,26,28], and (b) the deep learning CNN technique [89], for the classification of three arbitrarily selected datasets (Indian Pine, Botswana, and Barrax) to illustrate the effectiveness of BS that utilizes only spectral information. The results were the average of five repeated runs, and the training data was 10% throughout. It is seen that both figures exhibit a similar trend of behavior: the classification accuracy is seen to improve steadily when a greater number of bands are utilized for the classification until all of the bands of the imagery are exhausted. It is seen that the peak accuracy is saturated at the point when all the bands have been used for the classification. A similar trend has been seen over the many BS papers reported in the literature [2,3,4,5,6,7,8,9,10,11,12,13], which makes one speculate why the well-known bell shape of the accuracy-dimensionality curve predicted by Hughes [1], has not been observed from any experiments so far. Hughes analysis has shown that the theoretical accuracy of a model scales non-linearly with the dimensionality of the dataset: the accuracy should be improved when more spectral bands are utilized for the classification, and furthermore, increasing the dimensionality of the data for classification reduces the accuracy, especially when the training data size is kept constant. 

The experimental results as shown in Figure 2 were performed by using a fixed training data size of 10% per class throughout the experiment. Hence, there are two possibilities that may cause the bell-shaped accuracy-dimensionality curve not observable from Figure 2: (i) the number of training data that have been used in this experiment (10% training data) may be abundant enough for classifying 200 bands of data already; and (ii) the BS schemes adopted for this experiment are not effective enough to reach the theoretical peak of accuracy which should occur when a moderate number of bands are used for the classification. To testify the validation of the first point, Figure 3 plots the overall accuracy (OA) of the CNN-based algorithm for the experiments that have employed reduced training data of 5% and 3%, as shown in Figure 3a,b, respectively. It is seen that both plots exhibit the same trend as that depicted in Figure 2, and these accuracy plots do not seem to saturate even when all bands have been used for the classification. We have repeated the same experiment for 7 other BS algorithms (see Section 2 for more details of these competing BS schemes) and have observed the same trend of results, which will be presented in Section 3.2. These data may suggest that the absence of the Hughes′ accuracy-dimensionality classification characteristic in Figure 2 and Figure 3 is not due to the excessive amount of training data that has been utilized for the classification. The next step is to evaluate the efficiency of the BS schemes in an attempt to understand whether it may be the cause for the absence of the Hughes′ phenomenon in the present results. Note that the KNN classification results exhibit the same trend as those shown in Figure 2 and Figure 3, but they are not presented here for clarity.

### 3.2. Band Selection (BS) Using Spatial and Spectral Information

As outlined in Section 2 (Method and Materials), that spatial information has been regarded as an added advantage for the remote sensing data analysis, especially in applications related to ground-use classification. This section is devolved to the understanding of the result obtained from the previous Section 3.1 and with a view to giving more insight into the query of why the Hughes′ accuracy-dimensionality characteristic has hardly been observed from the present results, and neither has it been ever reported in the remote sensing literature. In the previous Section 3.1, it is suggested that the effectiveness of the BS may be one of the issues responsible for observing the results shown in Figure 2 and Figure 3. Thus, in this section, other techniques such as the use of spatial features for improving the effectiveness of band selections, and subsequently the enhancement of classification efficiency of hyperspectral data analysis, are studied here.

As mentioned in Section 2.2, a basic BS scheme that utilizes the mutual information (MI) for the selection of bands from hyperspectral datasets has been adopted in this study. The algorithm has been a basic one that ranks the MI of each spectral band of the dataset, and then they are selected from the top of the list for classification. Since this BS method utilizes spectral information only, a preprocessing method that exploits spatial information for the elimination of low discrimination bands has been added as a preprocessing technique prior to the MI band selection. This method is termed as ‘spatial spectral mutual information (SSMI)’ (see Section 2.3), and the sole purpose of the spatial technique is to eliminate bands that do not convey much information toward the morphological property of the dataset for classification. Figure 4 plots the ranked correlation coefficients C_i_ of Equation (2), and the abrupt change of C_i_ is detected as the threshold for band elimination. The abrupt change of the slow-varying Ci has been implemented by moving point smoothing (typically 11 points) of the vector; then, the breakpoint is detected by using the Matlab command ‘findchangepts’. Then, the remaining bands are subsequently processed by the basic MI scheme as detailed in Section 2.2. The effectiveness of the MI and the proposed SSMI for the classification of a couple well-studied datasets (Indian Pines and Botswana) is shown in Figure 5.

The classification accuracy is seen to reach a peak at about 20 bands when the data are treated by the proposed SSMI, and then, the accuracy is reduced steadily after the peak, when more bands are added to it. This behavior confirms to the Hughes’ prediction, and it is believed that this result may represent the first experimental evidence to confirm the validation of Hughes’ theory. Similar to the results presented in Figure 2 and Figure 3, the training data that have been utilized in this experiment is also 10% per class for the classification of both datasets (Indian Pines and Botswana). It is also noted that the classification performance of the MI BS scheme, which is shown in red trace in Figure 4 and Figure 5, is completely different from that of the SSMI result (in blue trace). Since both methods, the MI and the proposed SSMI, are fundamentally the same algorithms, it is interesting to find out why the SSMI exhibits such dramatic results with an enhancement of classification by about 12–15% better when a small number of bands (e.g., at approximately 20 bands) have been used for classification. Figure 6 plots the I(A, B) (see Equation (5) of the Indian Pines scene, which reveals that the I(A, B) of bands that are processed through SSMI (i.e., after the spatial treatment) have approximately 2 times higher contrasts (i.e., the peak and valley) than those processed by the MI. This may be one of the reasons why SSMI performs so much better than the MI counterpart even though the underlying principles of both techniques are literally the same. As an example, the red and black solid square markers shown in Figure 6 depict the bands that have been chosen by the MI and the SSMI respectively, when the dimensionality is set to 5 (bands). One significant difference between these 2 sets of selected bands is that the MI method chooses band 86 instead of the higher MI bands between 110 and 140. The consequence of this ‘erratic’ band selection is the drop of the OA by approximately 15% with respect to the SSIM, which scores an OA of 83% (see Figure 5a). The credit of this improvement is solely due to the elimination of the low morphological (non-discriminative) bands through the spatial feature analysis.

The robustness of the proposed SSMI is further testified by examining its band selection capability with respect to seven other competing algorithms (see Section 2 for details): (a) Saliency, (b) HyperBS, (c) SLN, (d) OCF, (e) FDPC, (f) ISSC, and (g) CNN. All experiments were conducted under the same configurations of (i) 10% per class of training data, (ii) experiments were repeated 5 times, and (iii) all experiments were classified by the same classifier (SVM). Again, the KNN results have been omitted here for clarity. The classification results of six HSI datasets, namely the Botswana, Indian Pines, Barrax, KSC, Salinas, and Pavia University, are presented in Figure 7. It is seen from the figure that the proposed SSMI achieves the best performance over all 7 competing algorithms, with a peaked classification accuracy at about 20 selected bands over all six datasets. None of the 7 competing algorithms exhibit a ‘bell’-shaped accuracy-dimensionality characteristic curve, except for the proposed SSMI method and the ISSC, which also exhibits a weak ‘bell’ shape (see Figure 7c). At the selected bands of about 20, the SSMI BS scheme achieves an enhanced averaged accuracy with respect to the mean of the 7 competing algorithms over 6 datasets that is approximately 10.5% better than all the competing algorithms employed in this study. This result may give evidence that the ‘knee’ shape of the accuracy-dimensionality curve generally seen from the BS schemes published in the literature may be predominately caused by the inefficiency of the reported BS algorithm. Further work along this line of research will be reported in the forthcoming publication. We have obtained a similar trend of results when the Kappa coefficients are used for assessing the goodness of the SSMI band selection scheme. To maintain the clarity of the paper, the plot of the Kappa coefficients is not presented here.

It is noted from Figure 7 that the peak classification accuracies for all datasets are seen to occur at about the same number of bands (i.e., at approximately 20 bands), despite the rather distinct different characteristics among the datasets that have been employed in this study. For example, the Pavia dataset contains 4 classes of manmade materials that are quite different in the spectral perspective from that of the natural vegetation scene in the rest of the datasets. However, according to the Hughes analysis, it is indicated that the peak of the classification accuracy is a function of the measurement complexity (i.e., the dimensionality of the data) as well as the number of the training sample that is required to define a class in the dataset [1]. Hughes′ analysis is valid only when the datasets concerned are noiseless, have a minimum of subpixel mixing, and also all spectral bands carry the same extent of information. Through the proposed SSMI BS method, all the datasets that are employed in this study have been treated such that the noisy bands and also those that are not rich in information have been put to the bottom of the band selection list. All other bands have been ranked in the order of the information contained in the band; thereby, the relative difference of the band information for any band in the list with respect to the most informative ones (i.e., the one at the top list) is more or less similar over all the 6 datasets. This may be one of the contributive factors for the observation of the peak accuracy at approximately 20 bands over all the datasets, as depicted in Figure 7, and more detailed investigation for further understanding this query is in progress.

The Hughes analysis indicates that the peak accuracy varies with the number of the training sample. The OA of all datasets that have been classified by SVM using 10% and 3% training data are shown in Figure 8a,b respectively. It is clear that the centers of the peak accuracies for the classification using 3% training data (Figure 8b) have been shifted rather significantly with respect to that of the 10% training. The shift of the peak accuracy can be seen better in Figure 8c,d, where the OA of the Pavia and Indian Pines datasets classified by SVM (and KNN) using the 10% and 3% training data are overlaid together for better visualization of the shift of the classification accuracy peak. The shift of the peak in the Indian Pines data set (Figure 8d) is consistent with all the subplots presented in this figure, and it is noted that there is a subtle difference between the SVM and the KNN result.

To understand further how the SSMI enhances the classification accuracy, Figure 9 and Figure 10 display the false color classification maps of 3 randomly chosen datasets (Indian Pines, Pavia University, and the Salinas), which have been processed through the MI and the proposed SSMI BS schemes. The presented classification maps are the results for the selected bands of 20 where the peak of the accuracy occurs. Figure 9 displays the classification result of the complete scene, while the zoom-in of the classification maps that highlights the enhanced classification ability of the SSMI is presented in Figure 10. It is quite clear from both figures that the classification capability of the SSMI is over 10% (see Figure 5 above) better than the counterpart MI algorithm, which only utilizes spectral information for band selection.

## 4. Discussions

According to the results presented in the last section, it is clear that the elimination of the non-discriminative bands are essentially important for enhancing the classification accuracy. It is also observed that not only the elimination of counter-productive bands is critical, the method for the selection from the ensemble of informative bands is also important, too. As an example, Figure 11a plots three trials of band selections schemes (for 5 bands) for the classification of the Indian Pine data, and the corresponding effects of the selections of these bands to the OA are shown in Figure 11b. The square marker dictates where the bands are taken from: the green, red, and black square markers which represent the bands that have been selected by the MI (i.e., the spectral BS scheme), the bands as according to the trial set (#1) and the trial set (#2), respectively. The trial sets #1 and #2 are two manually modified band sets with an objective to monitor the effects of replacing or omitting particular bands to the OA when the classification is performed by using these trial sets of bands. As discussed in Section 3, the bands that are selected by the MI BS scheme (depicted by green markers in Figure 11a) have the lowest OA (at 72% for the selection of 5 bands), which is due mainly to the selection of band 86, which has the lower I(A,B) than the other 5 main band clusters. The inclusion of band 128 in set (#1) (red square marker) increases the OA by approximately 7% w.r.t. that using the selected bands by the MI BS scheme. Furthermore, the inclusion of the band 71 in set (#2) (black square marker) increases the OA by approximately 12.5% over that of the MI BS result. This is surprised to observe the significant influence of the OA as much as >10% by the inclusion or missing out of a single band. Thus, these data further support the result of the previous section that more work is needed to study how the efficiency of the band selection scheme can be optimized.

## 5. Conclusions

One of the main objectives of this paper is to study why the ‘curse of dimensionality’ (or so-called Hughes’ phenomenon) [1], has not been observed experimentally so far, despite numerous reports on the subject of band selection (BS) in the hyperspectral imaging (HSI) analysis. In the literature, the accuracy-dimensionality curves are commonly reported in the form of a ‘knee’ shape, instead of the theoretically predicted ‘bell’-shaped characteristic. The query of why the theoretical prediction has not been observed remains to be an open question till now. Possible answers have been prescribed: (i) the number of training data is abundant enough to train the high degree of freedom of the classification model, or (ii) the reported BS schemes are not efficient enough to reveal the intrinsic (i.e., theoretical) accuracy-dimensionality characteristics.

To address the first factor, a series of experiments that utilized a successive reduction of training data, in the range of 10%, 5%, and 3% for the classification of datasets with small class sizes (e.g., Botswana) have been performed. The accuracy-dimensionality curves of these experiments have been unchanged and they remain in ‘knee’-shape forms. This shows that it is not the training data that have caused the Hughes’ theoretical prediction to be unobservable in these experiments. 

To study the second possibility, a band selection (BS) scheme that utilizes both spatial and spectral information for enhancing the classification accuracy of hyperspectral imagery (HSI) has been designed. The BS scheme utilizes a spatial feature extraction as a preprocessing step, followed by a basic mutual information (MI) spectral feature-based BS method, which is known as the spatial spectral mutual information (SSMI) scheme. The classification result has revealed that the accuracy-dimensionality characteristic of the basic MI BS always exhibits a ‘knee’ curve that is independent of the amount of training data. In contrast, the classification through the enhanced SSMI BS scheme always shows a sharp ‘bell’-shaped accuracy-dimensionality curve with a peak at the dimensionality of about 20 bands. Then, the experiment is repeated for 6 HSI datasets (Indian Pines, Botswana, Barrax, Pavia University, Salinas, and KSC) to compare them with 7 other state-of-the-art BS schemes (Saliency, HyperBS, SLN, OCF, FDPC, ISSC, and CNN). In all cases, the experiments were conducted under 10% training data and the SVM (and KNN) classifiers have been employed for classification. The accuracy-dimensionality characteristic of all 7 BS schemes exhibit the same ‘knee’ shape, and only the proposed SSMI method reveals a ‘bell’-shaped accuracy-dimensionality curve that features a peaked accuracy at about 20 bands. At the peak, the enhancement of the accuracy is approximately 10% better than all 7 BS algorithms over 6 datasets that have been employed in this study. Based on this result, one likely answer of why the Hughes′ phenomenon is only observable from the proposed SSMI may well be due to the enhancement of the classification accuracy through the better efficiency of the SSMI BS scheme.

A further experiment has indicated that the classification accuracy can be affected as significant at approximately 10%, when a single band is included or missed out for classification. The present result has pointed out one key issue for the future research in BS: how can the efficiency of band selection be optimized, and what assessment metric should be used for the indication of the efficiency of band selection? It is obvious from the present study that the incremental improvement of classification accuracy that has been conventionally adopted for the indication of the goodness of band selection algorithm is not sufficient enough to reveal the intrinsic integrity of the proposed band selection scheme.

## Figures and Tables

**Figure 1 jimaging-06-00087-f001:**
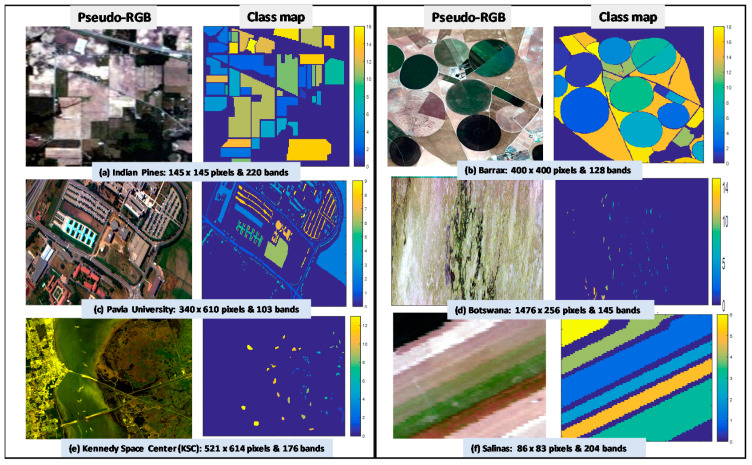
Shows the pseudo-RGB images (in the first and third columns) and their corresponding ground truth (GT) classification maps (in the second and fourth columns) for all the datasets utilized in this study: (**a**) Indian Pines, (**b**) Barrax, (**c**) Pavia University, (**d**) Botswana, (**e**) Kennedy Space Center (KSC) and (**f**) Salinas. Note that neither the dataset nor the GT has been modified such that the results presented in this work can be compared directly with those reported in the literature.

**Figure 2 jimaging-06-00087-f002:**
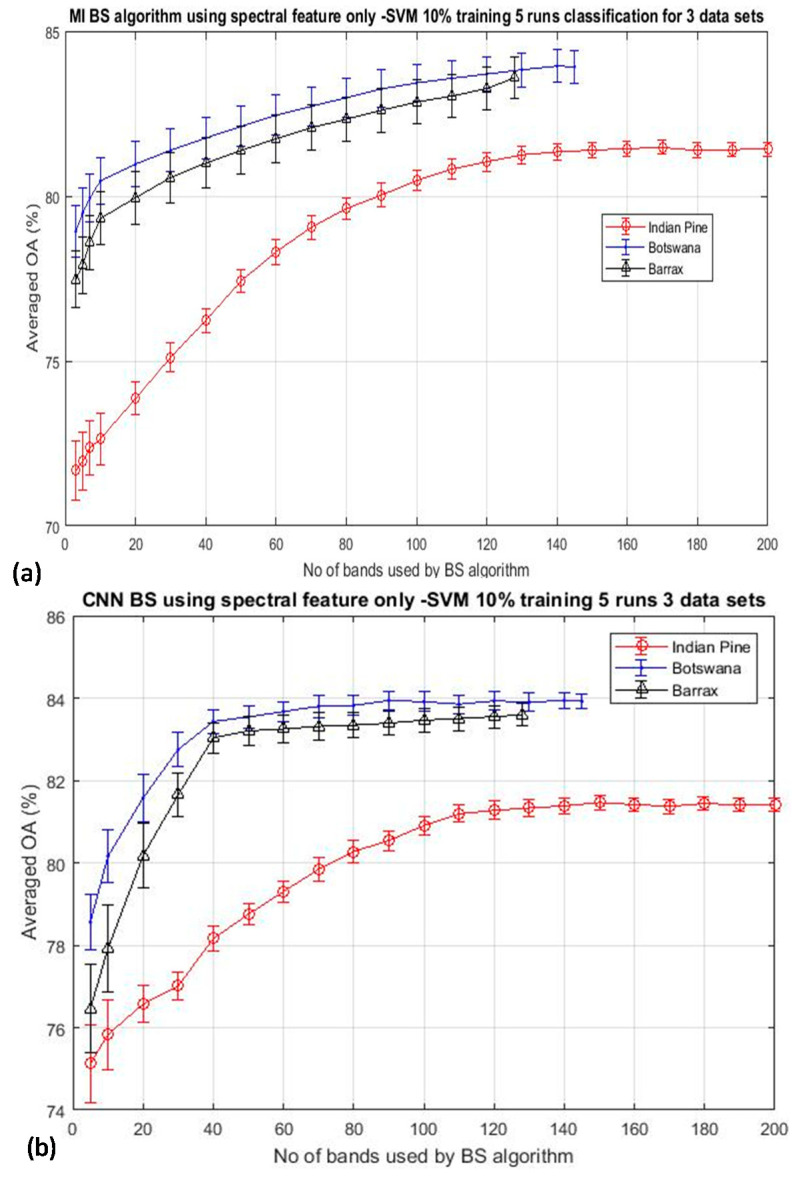
Shows the mean classification accuracy of two typical band selection (BS) algorithms as a function of the number of bands used for the classification of three datasets. In all cases, 10% of the training data per class have been used throughout all runs: (**a**) mutual information (MI)-based algorithm [17,26], (**b**) Deep learning CNN algorithm [89]. The continuous increasing of accuracy with the dimensionality makes one wonder why the Hughes’ accuracy-dimensionality phenomenon has not been observed here.

**Figure 3 jimaging-06-00087-f003:**
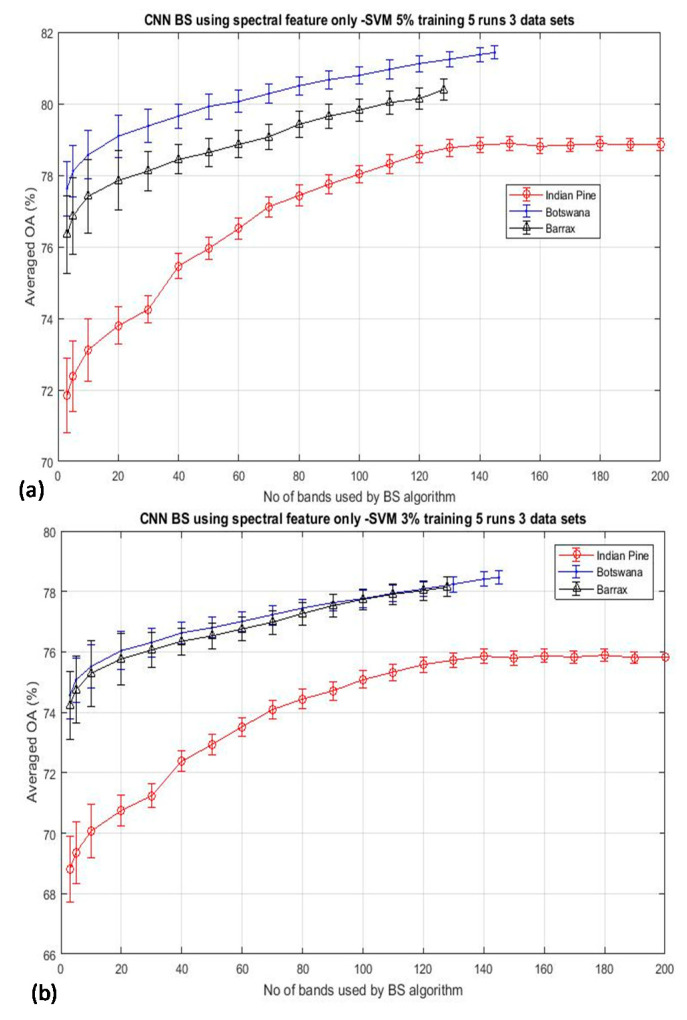
The mean classification accuracy of the Convolutional Neural Network (CNN) BS algorithms as a function of the number of bands used for the classification of the same datasets as in Figure 2 but with reduced training data to (**a**) 5% per class and (**b**) 3% per class. The trend of increasing the accuracy with the dimensionality under such a small amount of training data may suggest that it is not the ‘over-sufficient’ amount of training data that has been utilized here for causing the absence of the Hughes’ accuracy-dimensionality characteristic in these experiments.

**Figure 4 jimaging-06-00087-f004:**
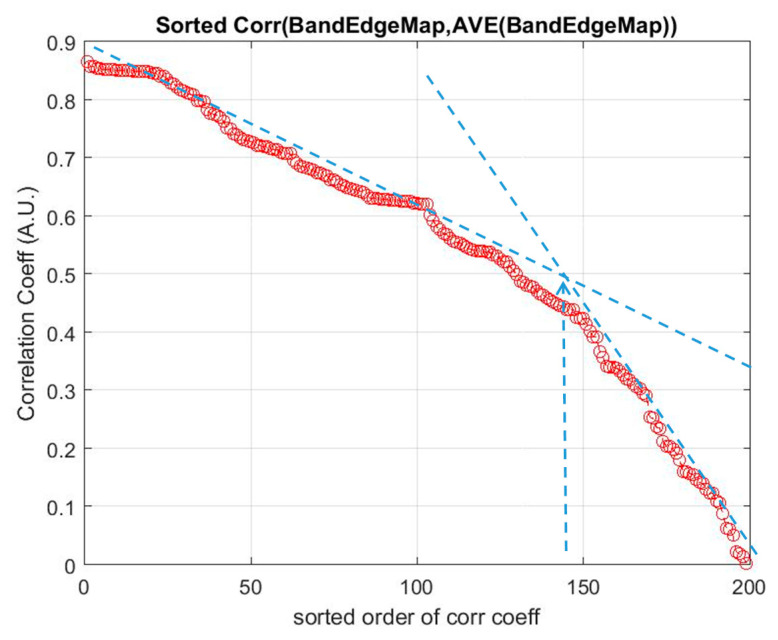
The spatial preprocessing technique (see Section 2.3) eliminates the non-discriminative bands through the abrupt change (blue arrow) of the correlation coefficients C_i_ of the edge map of each band, w.r.t.: The mean of all the edge maps in the dataset (Indian Pines). The abrupt change of the ranked C_i_ is detected automatically, and all bands beyond this point are discarded.

**Figure 5 jimaging-06-00087-f005:**
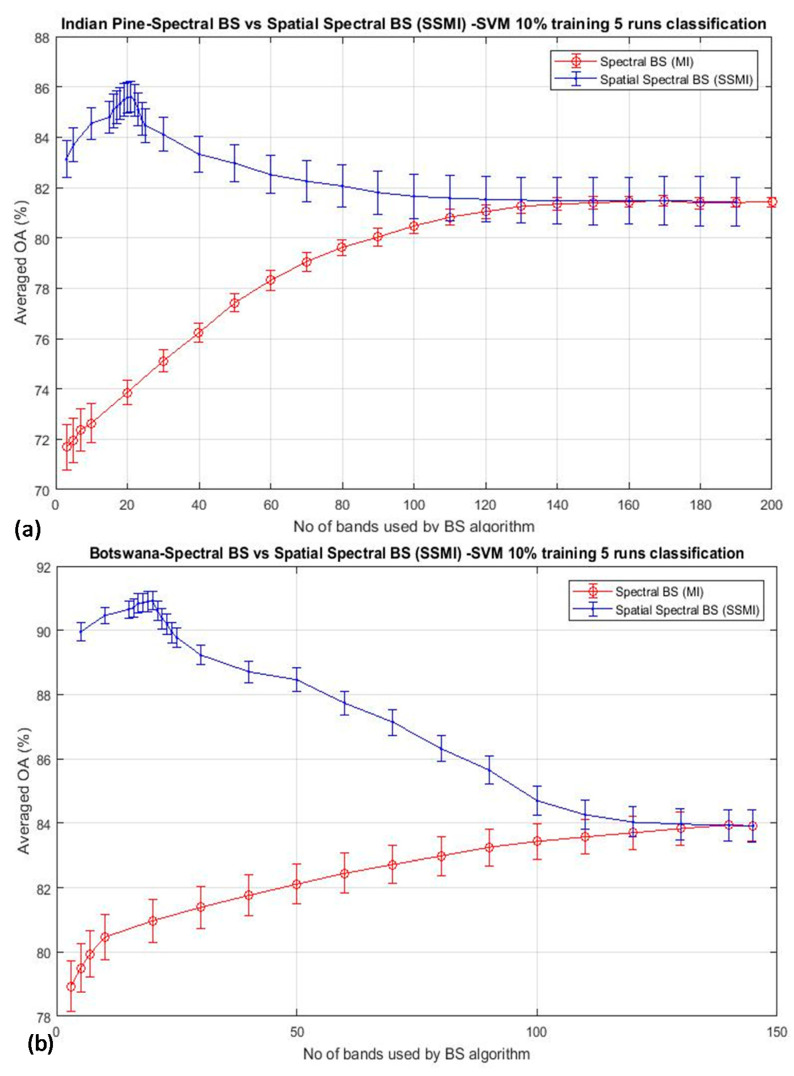
The mean classification accuracy of two BS algorithms: the MI and the proposed spatial spectral mutual information (SSMI), as function of the number of bands used for the classification: (**a**) Indian Pine and (**b**) Botswana datasets. In both cases, a ‘bell’-shaped accuracy-dimensionality curve similar to that predicted by Hugh [1], is seen, for the first time, from the experimental result of the proposed SSMI scheme (in blue plot). This is in great contrast to the basic MI algorithm (in red plot), which increases the accuracy steadily when more bands are utilized for classification.

**Figure 6 jimaging-06-00087-f006:**
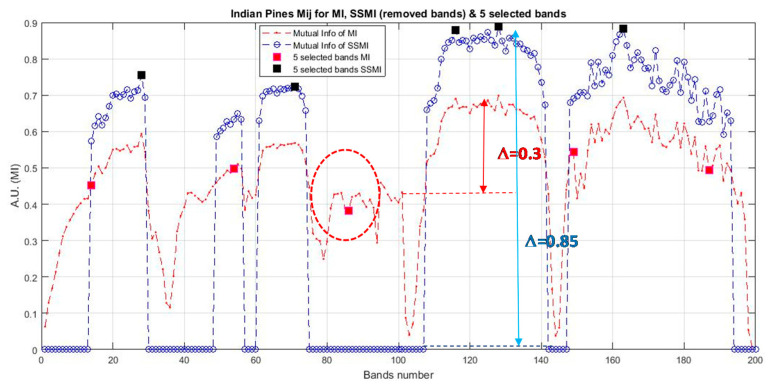
Depicts the I(A, B) of the bands of the Indian Pines dataset given by the MI (red plot) and SSMI (blue plot) BS methods. Note that the SSMI that utilizes both spatial and spectral information enhances the contrast of the I(A, B) by as much as approximately 200% in comparison to the basic MI BS without spatial preprocessing.

**Figure 7 jimaging-06-00087-f007:**
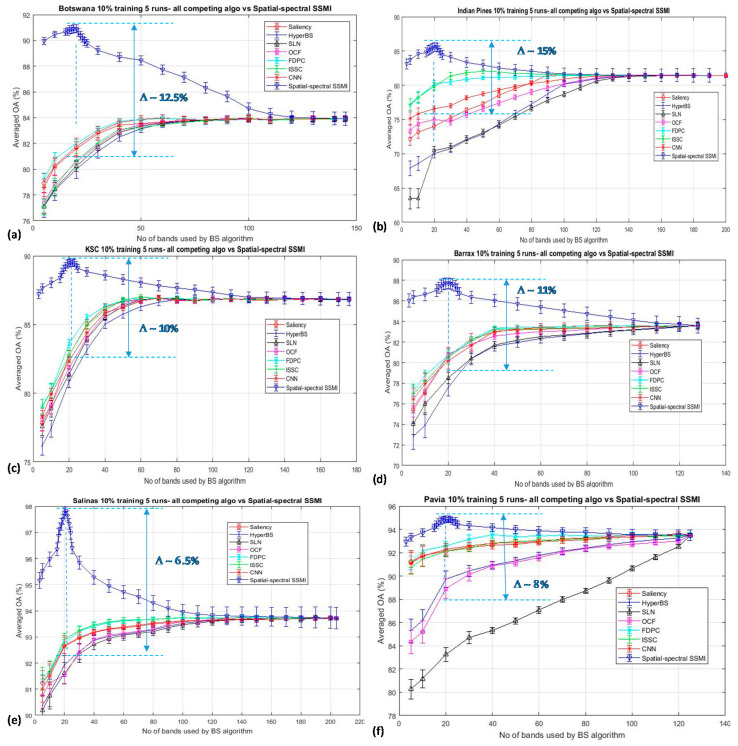
The classification performances of the proposed SSMI and seven other competing algorithms for the classification of 6 datasets (**a**) Botswana, (**b**) Indian Pines, (**c**) KSC, (**d**) Barrax, (**e**) Salinas, and (**f**) Pavia University. In all cases, the training data sizes were kept at 10% per class and every data point involves 5 repeated experimental runs. The averaged enhancement of the classification by the proposed SSMI with respect to the mean of the 7 competing algorithms over 6 datasets at the selected bands of 20 is approximately 10.5%.

**Figure 8 jimaging-06-00087-f008:**
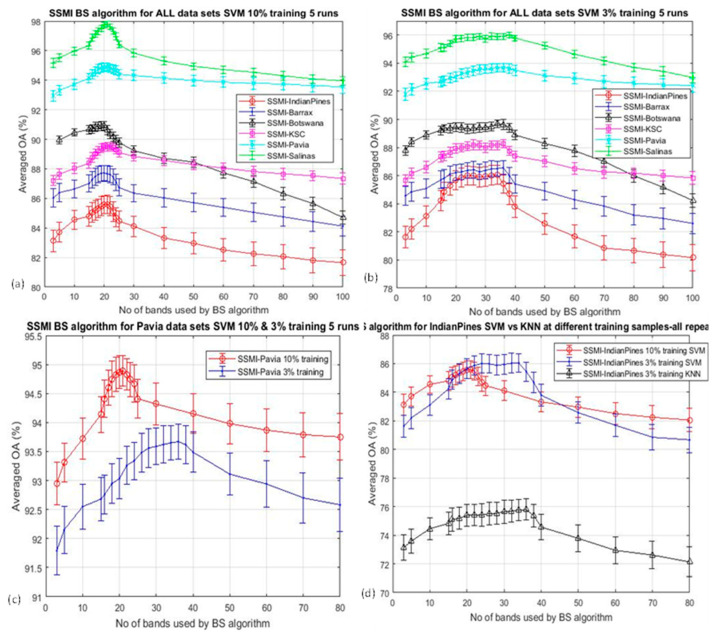
Plots the overall accuracy (OA) for the classification of all six datasets after band selections using the proposed SSMI algorithm: (**a**). SVM under 10% training data; (**b**). SVM under 3% training data; (**c**). the overlay of OA for the classification of the Pavia dataset by SVM using 10% and 3% training data; (**d**) as in (**c**) but for the Indian Pines datasets and also to compare with that by using the KNN classifier at 3% training data.

**Figure 9 jimaging-06-00087-f009:**
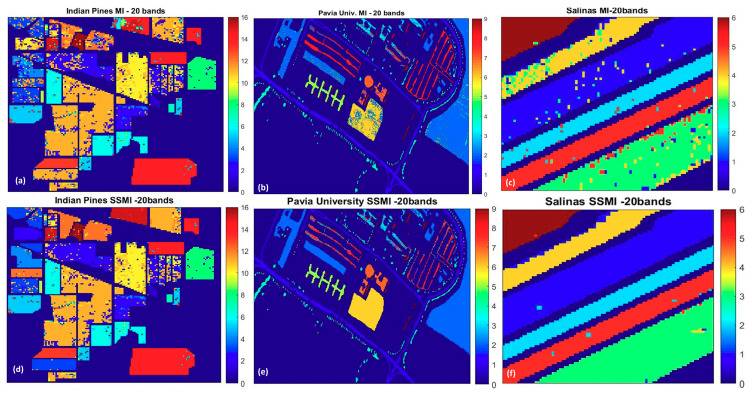
The false color classification maps of three datasets: (**a**,**d**): Indian Pines, (**b**,**e**): Pavia University, (**c**,**f**): Salinas; obtained by SVM classification (10% training) through the MI BS scheme (Upper panel) and the proposed SSMI method (Lower panel), which exhibits a substantial reduction of false alarms.

**Figure 10 jimaging-06-00087-f010:**
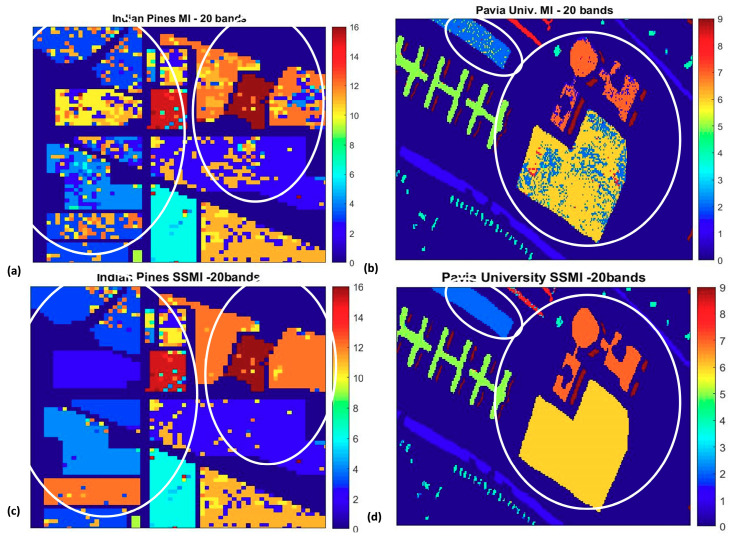
The zoom-ins of the false color classification maps in Figure 9 to highlight the substantial reduction of classification false alarms in the classes (circled) when the bands are selected through the SSMI BS method. Left column (**a**,**c**): Portion of Indian Pine, Right column (**b**,**d**): Pavia University dataset. The classification results using bands selected by the MI BS (Upper panel) and those proposed by the SSMI BS scheme (Lower panel).

**Figure 11 jimaging-06-00087-f011:**
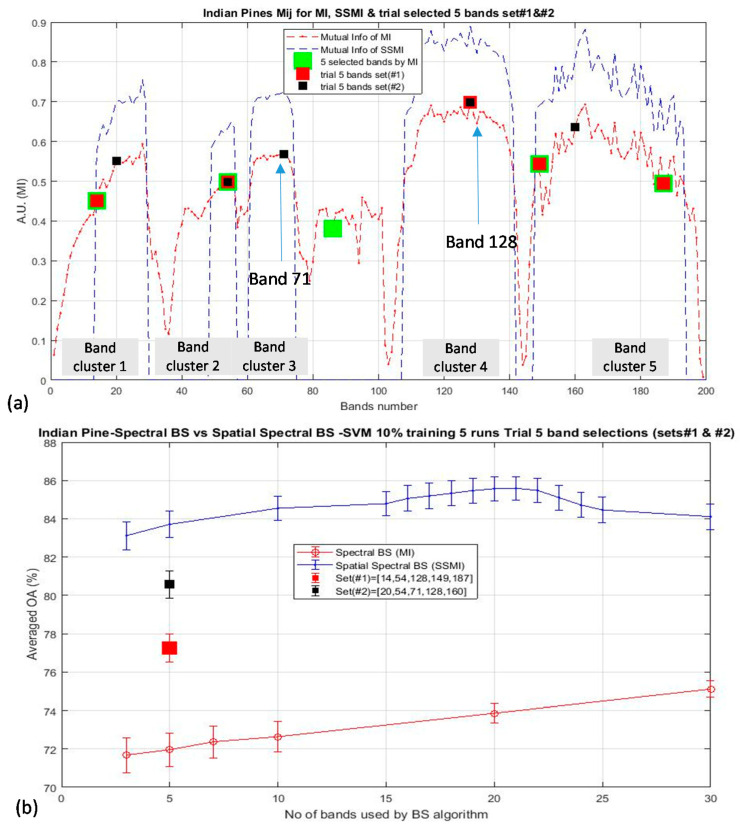
Demonstrates the significant influence of the classification accuracy of the Indian Pines data by the inclusion or missing of crucial bands: (**a**) Illustrate the 5 board clusters of I(A, B) and to indicate which bands have been selected or deselected according to the 3 selection schemes of (i) by MI BS algorithm (in green marker), (ii). Manual trial set #1 (in red marker), (iii). Manual trial set #2 (in black marker). (**b**) Illustrates the significant influence of the OA (as much as 10%) when one ‘crucial’ band is selected for classification at the dimensionality of 5 band (see text for details).

**Table 1 jimaging-06-00087-t001:** Pseudo-code of the proposed spatial spectral mutual information (SSMI) method.

**Algorithm 1: Spatial Spectral Mutual Information (SSMI)**
**Input: Im = (x,y,B), threshold;** **Output: MI**
% In Matlab format %
%% Spatial preprocessing %%
B = (B_1_…B_N_)
E = Edge (Im(B_i_ ,… ,B_N_))
mE = mean(E)
Ci = Corr(E,mE)
CiRank = sort(Ci,’descent’)
%either manual or automatic threshold
CiSelect = CiRank(1:threshold)
%% Spectral band selection
ImSpec = Im(x,y,CiSelect)
S = size(CiSelect)
% Joint entropy evaluation
For i = 1: S
%choose adjacent image pair
Impair(i) = [ImSpec(:,:,i),ImSpec(:,:,i+1)] %normalise joint histogram
H(i) = Impair(i)/sum(sum(Impair(i)))
%joint entropy
JE(i) = −sum(H(i).*(log2(H(i))));
MI(i) = (entropy(Im(:,:,i))+entropy(Im(:,:,i+1)) − JE(i))/JE(i)
end

**Table 2 jimaging-06-00087-t002:** Tabulates the class sizes and the nature of the class for all six datasets that have been utilized in this study. It is to note that the class sizes of the Botswana dataset are relatively uniform with an average size of 237 pixels over all classes in this data set.

Class	Pavia University		Indian Pines		Barrax		Salinas		KSC		Botswana	
	Class Label	Number of Samples	Class Label	Number of Samples	Class Label	Number of Samples	Class Label	Number of Samples	Class Label	Number of Samples	Class Label	Number of Samples
**1**	Asphalt	6631	Alfalfa	46	Alfalfa	20606	Brocoli Green weeds_1	391	Scrub	875	Water	270
**2**	Meadows	18649	Corn-not ill	1428	Corn (two leaves)	13839	Corn_senesced green_weeds	1343	Willow swamp	279	Hippo grass	101
**3**	Gravel	2099	Corn mint ill	830	Corn (five leaves)	4921	Lettuce_romaine 4wk	616	Cabbage palm hammock	294	Floodplain grasses 1	251
**4**	Trees	3064	Corn	237	Corn (six leaves)	2063	Lettuce_romaine 5wk	1525	Cabbage palm/oak hammock	290	Floodplain grasses 2	215
**5**	Painted metal sheets	1345	Grass-pasture	483	Beet	5496	Lettuce_romaine 6wk	674	Slash pine	185	Reeds 1	269
**6**	Bare soil	5029	Grass-trees	730	Legumes	298	Lettuce_romaine 7wk	799	Oak/broad leaf hammock	263	Riparian	269
**7**	Bitumen	1330	Grass-pasture-mowed	28	Wheat	11554			Hardwood swamp	121	Fire scar 2	259
**8**	Self-blocking bricks	3682	Hay-windrowed	478	Experimental plots (legumes)	4965			Graminoid marsh	496	Island interior	203
**9**	Shadows	947	Oats	20	Experimental plots (papaver)	5118			Spartina marsh	598	Acacia woodlands	314
**10**			Soybean-not ill	972	Lignose	1972			Cattail marsh	465	Acacia shrublands	248
**11**			Soybean mint ill	2455	Vineyard	949			Salt marsh	482	Acacia grasslands	305
**12**			Soybean-clean	593	Test plots	3245			Mud flats	578	Short mopane	181
**13**			Wheat	205	Lysimeter station	534			Water	1066	Mixed mopane	268
**14**			Woods	1265	Water body site	62					Exposed soils	95
**15**			Buildings-Grass-Trees-Drives	386	Non-irrigated barley	26132						
**16**			Stone-Steel-Towers	93	Irrigated barley	976						
**17**					Bare soil	11357						
**18**					Ploughed soil	1196						

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
