# Peer review of "Spatial Spectral Band Selection for Enhanced Hyperspectral Remote Sensing Classification Applications"

_2313-433X, 2020, doi:10.3390/jimaging6090087_

Round 1

Reviewer 1 Report

This study presents a new classification scheme, SSMI, which first orders bands by spatial information content, and then calculates the mutual information of these ordered bands to determine bands with the highest spatial and spectral information content. This improved band selection results in increased classification accuracy when compared with other classification methods. 

The paper was interesting and well laid-out, although I have a significant concern that the peak in classification accuracy happened after 20 bands for every one of six (very different) images under consideration (see Figure 6). Why do you think that number would be the same for all images? It certainly makes me doubt the representativeness of the method.

Hughes' phenomenon is referenced throughout, and seems to be one of the major justifications of the paper. If the authors reframed the paper in terms of classification accuracy then it would be an interesting incremental addition to the classification literature. But dealing with Hughes' paradox requires more work, I believe. It was my understanding that a peak in classification accuracy is not typically seen in HSI due to sub-pixel mixtures and the presence of correlated spectral noise. Part of the re-ordering of the data in the spatial pre-processing step would then essentially break the spectral correlation of the noise. This sort of effect would be best studied by constructing simulated datasets using spectra from libraries in known spatial patterns, and evaluating the impact of different class numbers/sizes, correlated/uncorrelated noise of different levels, etc. Showing a change in the peak location using those simulated images would also alleviate my concern regarding the consistent peak in Figure 6. 

Some more minor suggestions are given below:

  • P.3 L.115 and Section 3.2. How do you determine the "abrupt change" for the subsetting of bands in the pre-processing step. You say it's automated, but how does that automation work?
  • P.3 L.134. Earlier you used subscript i for bands; I suggest you use a different subscript to indicate pixels.
  • P.4 L.155. "normalized histograms" implies some sort of binning. It may be better to just refer to "normalized band pairs".
  • P.4 L.161. The way MI is described in the algorithm isn't well described in text.
  • P.6 L242. How were the ground classification maps derived?
  • P.6 L242 You say that the class sizes in Botswana are fairly uniform, however, the image seems to be dominated by a single class?
  • P.7 L254 In Table 2, Acacia woodlands appears twice for Botswana. Is this a duplicate entry?
  • P.12 L364-368 I believe the difference is largely due to ordering. Different band pairs are considered by each method.
  • P.15 L.435 This paragraph needs to be rewritten for clarity. A and B are variable names you've used before. I suggest new ones here to avoid confusion. How did you choose A and B? Why is the figure shown for MI and not SSMI?

Reviewer 2 Report

The article continues the old discussion on patterns classification starting from well known paper of GORDON P. HUGHES On the Mean Accuracy of Statistical Pattern Recognizers  (see also the comments) “On the Mean Accuracy of Statistical Pattern Recognizers” K. Abend and T. J. Harley, Jr.

Unfortunately, this problem could not have the precise mathematical solution without exact knowledge what kind of features are necessary for recognition. Anyway the attempt to characterize these features in terms of spatial resolution could be useful. Anyway in aerial photography and in terrain based recognition systems in TV or IR range, particularly in military systems there are special standards which permit to compare various observation systems. But these standards involves not only the typical images, but also the position of observation object, type of background, illumination, seasonality and many others. In the development procedures of the observation systems all these features and even many others should be taken into account, from my view point the spectral features are just one of possible approaches. Anyway, it would be nice if the authors pay more attention to the multiple features besides the spectral, moreover the machine learning may be useful, but not a principal part of development of the observation system. As for the article itself it is full of general statements related the probability and information which do not have any sense without exact mathematical problem statement.

Reviewer 3 Report

This paper proposed a spatial spectral mutual information (SSMI) band selection (BS) scheme and experimentally investigated the absence of the “curse of dimensionality” of current BS methods. The results look very good especially when less spectral bands are selected. The extensive experiments on six datasets demonstrate the effectiveness of the proposed method. Generally, the paper is well written. Before acceptance, the following comments have to be addressed:

  1. According to the experimental results in Fig. 7c, some compared methods also show weak ‘bell’ curves such as the ISSC method. Thus, the discussion in line 417 “None of the 7 competing algorithms, exhibit a ‘bell’ shape …” has to be reformulated more precisely.
  2. The pre-processing step of removing some bands seems to be very important. The related thresholds in the experiments should be given. Also how to determine them is also encouraged to discuss.
  3. The results of the proposed method are very impressive especially when very limited number of bands are selected in comparison with the existing methods. If possible, could the authors provide me a Matlab script to reproduce the SSMI curve in Fig. 5 (a)? The script should include detailed information about which bands are selected (can be as input for the script) and how the selected bands are fed into SVM (accuracy is output).
  4. Typo: Line 384 conforms -> confirms

Reviewer 4 Report

I appreciate the opportunity of reviewing this paper, which I accept for publication with minor changes.

The only general comment that I want to mention is: the accuracy achieved by a classification model when applied to a specific problem is in general therms a function of how does the model is able to fit such problem. In remote sensing, various issues may influence the complexity of a classification problem: among which one can mention the number of classes, the similarity between classes, the amount of features available, the size of the training set, GSD size, etc. So the paper brings a interesting contribution but it does not finish the theme.

I have some punctual comments that I present in the following lines.

Abstract:

line 16: I would avoid citing inside the abstract. In my opinion, I would just remove it, since the ‘curse of dimensionality’ is a well-known and discussed phenomenon in the filed of pattern recognition. Another alternative may be providing the reference in full within the body of the abstract.

Line 28: It is written “Hugh’s”, but as far as I am concerned “ Hughes’ ” is the correct form. Please, check in the remainder of the manuscript, I say lines 28, 32, 43, 72, 309, 310, 320, 333, 336, 344, 350, 378, 384, 385, 486, 498 and 515.

Line 147: In the right side of equation 3 there is a “)” missing.

Lines 152-153: the number in the equation (4) is misplaced.

Lines 161-163: the number in the equation (5) is misplaced.

Line 186: It seems to be necessary including a page break before Table 1.

This algorithm presentation must be improved. It is written using a syntax too much dependent to the programming language (probably matlab, including semicolon to end a statement without echo). I would use a more abstract language (like natural math symbology as well as pseudocode), since it is more universal. I would say also to use indentation. Anyway, if you prefer keeping it in its original programming language, please, at least, remove all semicolons (at the end of a statement line), apply indentation, put comments before the referenced line (not side by side), and cite the language in which it was written.

Line 274: I believe the correct form of the text fragment “3 times small” should be “3 times smaller

Line 324: I am not quite sure but I have the impression there is a missing “not”, so the text “may be” would be “may not be”. Please, revise it to be sure it wrong or not

Line 348: Instead of “relate” I think it is “related”.

Line 390: I would remove the word both in such line.

Round 2

Reviewer 1 Report

Thank you for your responses to my suggestions. I remain unconvinced about the consistent location in the peak in classification accuracy.

From your response:

"However, the authors would like to draw the reviewer’s attentions that, the present result is centred around 6 publicly available HSI data sets which have been widely studied, but, none of them ever reported the accuracy-dimensionality classification characteristic resemble to that predicted by Hughes. In the contrary, a clear ‘hill-top’ accuracy-dimensionality characteristics have been observed from the same 6 data sets in the present study for the very first time."

  • I don't argue that the hill-top is not apparent. But it is not as significant an achievement if it represents training dependence and not accuracy as a function of number of bands used. 
  • If your 20 bands are correct, are the band locations the same in all 6 images? (In other words, are you stating that only 20 bands are necessary to acquire, as opposed to a full hyperspectral datacube?)

A few other comments on your revised manuscript:

  • p4. L155. "In band selection (BS) the band image can 156 be considered as A and the pixels in the band is Ai". As in my previous review, I find it confusing that you use Ai to represent pixels, when previously Ai represented band i. 
  • p8. L283. In your response you clarified that the standard deviations for Botswana were smaller than other images, but you did not explain why the Botswana image appears to show only a single class, while the table shows many classes of similar sizes.
  • p12. L357. You did not clarify the figure in any way after my first query. How do you determine the points at which to measure the slopes?

Reviewer 3 Report

Most of my comments are well responsed by the authors. However, the response to the most important one (comment 4) is not good. To be honest, I'm quite supervised that the standard svm classifier on Indian Pines can yield an accuracy of 82% with only 2 or 3 bands in the case of 10% training data. Compared with the results in the following papers, this is a huge improvement (around 20%). That's why I was requesting the matlab code to reproduce the results in Fig. 5 (a), ensuring that there is no mistake. Note that I'm not asking for the complete codes but the part to do svm classification with the selected bands. As svm is an open classification tool in machine learning, I don't think the authors will suffer from any loss.

[1] Q. Wang, et al. Optimal Clustering Framework for Hyperspectral Band Selection. (see Fig. 5 (a)), IEEE TGRS, 2018.

[2] W. Sun, et al. Fast and Latent Low-Rank Subspace Clustering for Hyperspectral Band Selection. (see Fig. 4 (a)), IEEE TGRS, 2019.

[3] A. Datta, et al. Combination of Clustering and Ranking Techniques for Unsupervised Band Selection of Hyperspectral Images. (see Fig. 5), IEEE JSTARS, 2015.

Round 3

Reviewer 1 Report

The author's review response shows an aversion to modification of the manuscript.
